# Design and protocol of a comprehensive multicentre biobank for abdominal aortic aneurysms

Hamid Jalalzadeh,[1] Reza Indrakusuma,[1] Jan D. Blankensteijn,[2] Willem Wisselink,[2] Kak K Yeung,[2] Jan H N Lindeman,[3] Jaap F Hamming,[3] Mark J W Koelemay,[1] Dink A Legemate,[1] Ron Balm[1]

► Additional material is published online only. To view please visit the journal online (http://dx.doi.org/10.1136/bmjopen-2018-028858).

HJ and RI contributed equally.

[1]Department of Surgery, Amsterdam UMC, University of Amsterdam, Amsterdam, The Netherlands
[2]Department of Surgery, Amsterdam UMC, Vrije Universiteit Amsterdam, Amsterdam, The Netherlands
[3]Department of Surgery, Leiden University Medical Center, Leiden, The Netherlands

**Correspondence to**
Professor Ron Balm;
r.balm@amsterdamumc.nl

## ABSTRACT

**Introduction** The pathophysiology and natural course of abdominal aortic aneurysms (AAAs) are insufficiently understood. In order to improve our understanding, it is imperative to carry out longitudinal research that combines biomarkers with clinical and imaging data measured over multiple time points. Therefore, a multicentre biobank, databank and imagebank has been established in the Netherlands: the '*Pearl Abdominal Aortic Aneurysm*' (AAA bank).

**Methods and analysis** The AAA bank is a prospective multicentre observational biobank, databank and imagebank of patients with an AAA. It is embedded within the framework of the Parelsnoer Institute, which facilitates uniform biobanking in all university medical centres (UMCs) in the Netherlands. The AAA bank has been initiated by the two UMCs of Amsterdam UMC and by Leiden University Medical Center. Participants will be followed during AAA follow-up. Clinical data are collected every patient contact. Three types of biomaterials are collected at baseline and during follow-up: blood (including DNA and RNA), urine and AAA tissue if open surgical repair is performed. Imaging data that are obtained as part of clinical care are stored in the imagebank. All data and biomaterials are processed and stored in a standardised manner. AAA growth will be based on multiple measurements and will be analysed with a repeated measures analysis. Potential associations between AAA growth and risk factors that are also measured on multiple time points can be assessed with multivariable mixed-effects models, while potential associations between AAA rupture and risk factors can be tested with a conditional dynamic prediction model with landmarking or with joint models in which linear mixed-effects models are combined with Cox regression.

**Ethics and dissemination** The AAA bank is approved by the Medical Ethics Board of the Amsterdam UMC (University of Amsterdam).

**Trial registration number** NCT03320408.

### Strengths and limitations of this study

► Longitudinal collection of clinical data, biomaterials and imaging data of patients with an abdominal aortic aneurysm provides ample opportunity to better understand the natural history and to search for prognostic risk factors that might benefit future treatment.
► The inclusion of patients with a small asymptomatic abdominal aortic aneurysm offers the possibility to study natural history from an early stage.
► Standardised collection and storage of biomaterials allows the analysis of patients included at different hospitals.
► Study follow-up is done at routine follow-up appointments to minimise participant burden yet also means that study follow-up will vary between patients.
► Patient recruitment currently only takes place in university medical centres that are tertiary referral centres.

## INTRODUCTION

An abdominal aortic aneurysm (AAA) is a focal dilatation of the abdominal aorta that affects mostly elderly men. The prevalence of AAA in the general population is 1.3%–2.2% in 65-year-old men.[1 2] An AAA is an asymptomatic disorder that is associated with a high risk of mortality in case of rupture.[3] Yet, the risk of rupture itself is difficult to measure accurately and also varies considerably between patients. Consequently, the management of patients with an asymptomatic AAA is focused on balancing the risk of rupture with other competing risks of death, with the aim of preserving quality and quantity of life. On the one hand, asymptomatic patients who are estimated to have a high risk of rupture, in general at a diameter of more than 5.5 cm for men, may be offered prophylactic AAA repair if the risk of rupture outweighs any procedural and/or competing risks.[4] On the other hand, asymptomatic patients for whom the risk of rupture is estimated to be smaller than procedural and/or competing risks will be offered surveillance, for example, those with an AAA diameter smaller than 5.5 cm or those with severe comorbidities. A large body of research has

been dedicated to determining the optimal surgical treatment for AAA, focusing on either the threshold diameter for repair, the method of repair or the outcome and follow-up after treatment.

Although many studies have tried to unravel AAA pathophysiology, this aspect is still insufficiently understood. Most of the current knowledge originates from histopathological studies that reflect the end stage of AAA disease. Early determinants and drivers of AAA formation are largely unidentified due to the unavailability of AAA tissue from an early stage of the disease. Therefore, the known determinants of AAA development are limited to general risk factors such as male sex, ageing, smoking or connective tissue diseases.[5 6]

Recent studies focused on biomarker research to find early disease markers and potential targets for pharmacological treatment. Promising circulating biomarkers included markers of matrix turnover (matrix metalloproteinases), markers of inflammation (interleukins and C reactive protein) and markers of lipid metabolism (lipoproteins).[7–9] Unfortunately, to date, none of these biomarkers have found their way to clinical practice. This is mostly due to their low prognostic value for AAA progression and because many studies have not corrected for factors such as smoking or comorbidities.[8] The same applies to new imaging biomarkers such as [18]F-fluoro-deoxy-glucose or biomechanical markers such as peak wall stress and wall shear stress.[10–12]

To better understand AAA pathophysiology and its natural course, it is imperative to combine studies with biomarkers, imaging markers and longitudinal data. To that end, a multicentre databank, biobank and imagebank has been established in the Netherlands: the '*Pearl Abdominal Aortic Aneurysm*', hereafter referred to as the AAA bank. The aim of this project is to facilitate future studies on AAA. It is part of the Dutch Parelsnoer Institute (PSI), which facilitates uniform biobanking in all eight university medical centres (UMCs) in the Netherlands.[13] The systematic collection of clinical data, biomaterials and imaging data will enable a diverse range of studies on patients with AAA. The AAA bank will especially focus on patients with a small AAA to collect longitudinal data early on in the development of AAA.

The first proposed study that will be carried out with the collected biomaterials is the 'Predicting aneurysm growth and rupture with longitudinal biomarkers' (PARIS) study. The PARIS study aims to determine the association between AAA progression and the evolution of serum levels of proteases and cytokines.

The scientific aims of the AAA bank are: (1) to gain insight in the pathophysiology and natural history of AAA; (2) to gain more knowledge about the rupture risk of AAA; and (3) to evaluate and improve treatment of patients with an AAA. Future studies with data from the AAA bank must adhere to these scientific aims.

## METHODS AND ANALYSIS

### Study design

The AAA bank is a prospective multicentre observational biobank, databank and imagebank of patients with AAA in The Netherlands (see table 1 for the WHO Trial Registration Data Set).

The active protocol at the time of writing is version 3, 22 December 2017. The AAA bank is embedded within the framework of PSI, which is cofinanced by the Dutch government and the Netherlands Federation of University Medical centres (NFU).[13] PSI was established in 2007 and aims to facilitate standardised nationwide biobanks and clinical databases.[14] At the time of writing, 17 different patient cohorts (*Pearls*) for different diseases have been initiated within the PSI framework. Examples include the Diabetes Pearl, the Pearl Neurodegenerative Diseases, the Stroke Pearl and the Dutch Pancreas biobank.[15–18] All these biobanks adhere to an internal regulatory framework, which prescribes legal and ethical rules concerning the conduct of all Pearl-related activities.[14]

The AAA bank has been initiated by the two UMCs from Amsterdam University Medical Centers (University of Amsterdam and Vrije Universiteit Amsterdam) and by Leiden University Medical Center. In accordance with the goals of PSI and NFU, the AAA bank aims to expand to the other Dutch UMCs. The AAA bank is currently an ongoing project, with active recruitment and collection of data and biomaterials. The first patient was recruited on 4 October 2017.

### Study population

All capable adults with AAA in participating university medical centres are eligible for inclusion in the AAA bank. This also includes patients who previously have undergone AAA repair. Patients who are incapable due to a ruptured AAA can also be included, using a special recruitment process, which will be described below. All included patients will be followed for as long as they visit their treating vascular surgeon.

### Recruitment procedure

Eligible patients are recruited at the inpatient or outpatient clinic of the department of vascular surgery of the participating hospitals by research physicians and/or data managers. Participants who agree to participate give written informed consent during their visit to the inpatient or outpatient clinic. When patients arrive in an emergency setting with a ruptured or symptomatic AAA, oral consent is required from either the patient or a legal representative. This oral consent has to be confirmed in writing at a later stage—either by the patient or by a representative in case of a fatal outcome. In the event that no written informed consent can be obtained, all data and biomaterials collected for the AAA bank will be destroyed.

| Table 1 | WHO trial registration data set |
|---|---|
| **Primary registry and trial identifying number** | **ClinicalTrials.gov: NCT03320408** |
| Date of registration in primary registry | 25 October 2017 |
| Secondary identifying numbers | NL59991.018.17, PARIS study, biobank Pearl AAA |
| Sources of monetary of material support | AMC Foundation for monetary support. |
| Primary sponsor | Academic Medical Center – University of Amsterdam |
| Secondary sponsor(s) | None |
| Contact for public queries | Els Kuiters, e.kuiters@amsterdamumc.nl,+31205667832, Meibergdreef 9, 1105AZ, Amsterdam, The Netherlands. |
| Contact for scientific queries | Principal investigator: Professor R Balm, r.balm@amsterdamumc.nl,+3120–5667832, Meibergdreef 9, 1105AZ, Amsterdam, The Netherlands. Els Kuiters, e.kuiters@amsterdamumc.nl,+31205667832, Meibergdreef 9, 1105AZ, Amsterdam, the Netherlands. |
| Public title | PARIS study and biobank Pearl AAA |
| Scientific title | Predicting aneurysm growth and rupture with longitudinal biomarkers (PARIS study) and biobank Pearl AAA. |
| Countries of recruitment | The Netherlands |
| Health condition(s) | Abdominal aortic aneurysm |
| Intervention(s) | None |
| Key inclusion and exclusion criteria | Inclusion criteria: adult with an AAA or who has been previously treated for an AAA. Adequate comprehension of the Dutch language to provide written informed consent. Exclusion criteria: decisionally impaired patients. The exception are patients who are decisionally impaired due to the effects of an acute AAA for whom a separate recruitment and consent procedure exists. |
| Study type | Observational longitudinal patient registry and biobank |
| Date of first enrolment | 4 October 2017 |
| Sample size | Planned: 750 Currently enrolled: 161 |
| Recruitment status | Recruiting; participants are currently being recruited and enrolled |
| Primary outcome(s) | Outcome: AAA growth. Time frame: up to 10 years of follow-up. Outcome. AAA rupture. Time frame: up to 10 years of follow-up. Outcome: all-cause mortality. Time frame: up to 10 years of follow-up. Outcome: evolution of serum levels of proteases and cytokines. Time frame: a maximum of 1 measurement annually up to 10 years of follow-up. Outcome: proteases and cytokine levels in AAA tissue. Time frame: if open AAA repair is performed and AAA tissue is collected. This is a one-time measurement. |
| Key secondary outcome(s) | Outcome: incidence and type of complications after AAA repair. Time frame: up to 10 years of follow-up after AAA repair. |
| Ethics review | Status: approved. Date of approval: 25 August 2017. Name and contact details of ethics committees: Medical Ethics Board of Amsterdam UMC (University of Amsterdam). mecamc@amc.uva.nl,+31205667389, Trinity building C, fourth floor, Pietersbergweg 17, 1105BM Amsterdam, The Netherlands. Biobank Ethics Board of Amsterdam UMC (University of Amsterdam), biobanktoetsing@amc.uva.nl,+=31205666730, Meibergdreef 9, 1105AZ, Amsterdam, The Netherlands. |
| Completion date | Expected: 4 October 2032 |
| Summary results | No results yet. |

Continued

| Table 1 | Continued |
|---|---|
| Primary registry and trial identifying number | ClinicalTrials.gov: NCT03320408 |
| IPD sharing statement | Plan to share IPD: yes.<br>Plan description: IPD sharing for research will be allowed in a data sharing procedure. Scientific requests need to fall under the scope of the scientific aims as formulated in this manuscript. Researchers willing to requests IPD can initiate this procedure by contacting the researchers. Data will be released depending on the scientific quality of the submitted request. |

AAA, abdominal aortic aneurysm; PARIS, Predicting aneurysm growth and rupture with longitudinal biomarkers.

## Study procedures

The study procedures of the AAA bank are embedded within regular AAA treatment and are carried out by physician-researchers and data managers. Standard operating procedures have been set up to minimise the amount of missing data. Clinical data and biomaterials are collected during and after visits that are already part of clinical treatment to limit participant burden and improve participant retention. No research visits will be planned for the AAA bank. In addition, only imaging data that is obtained as part of clinical care will be stored in the AAA bank.

In line with regular AAA management, four distinct study phases have been identified: inclusion phase, surveillance phase, surgical phase and postoperative phase (figure 1; blue boxes). Within each phase, multiple visits can take place—especially in the surveillance or postoperative phase. These phases can theoretically continue indefinitely, depending on the course of the AAA. With regard to the 'surveillance phase', the Dutch AAA guideline advises that patients visit a vascular surgeon at intervals of 2 years, 1 year or 3 months, depending on the AAA diameter.[19] Patients can move from the 'surveillance phase' to the 'surgical phase' and subsequently to the 'postoperative phase' (figure 1). Furthermore, there will be patients who at some point reach the threshold diameter for repair yet who do not undergo repair for various reasons such as severe comorbidity. In these cases, an individual clinical decision—unrelated to their participation with the AAA bank—will have to be made together with the patient whether surveillance continues with regular intervals or whether the patient chooses to quit with surveillance altogether. Patients who choose to continue with surveillance will still be asked for biomaterials and clinical data, while patients who quit surveillance can be analysed using the previously collected data up until that moment. Furthermore, the latter patients can also be included in survival analyses as mortality data can be sourced from either the municipal registry of persons or their general practitioner.

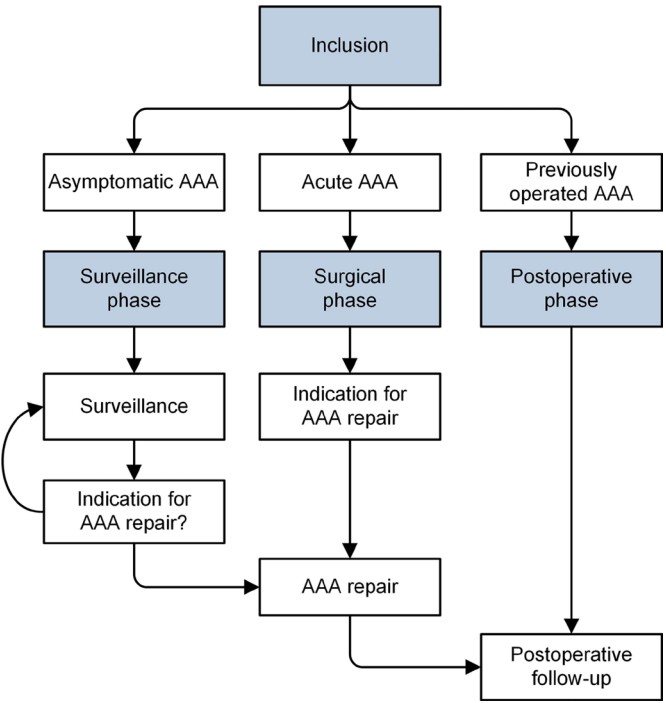

**Figure 1** Flow chart of study phases. AAA, abdominal aortic aneurysm.

## Clinical data: Parelsnoer Repository for Information Specification, Modelling and Architecture (PRISMA)

All clinical data are collected according to an information model called PRISMA, which was constructed with the assistance of an experienced information architect of PSI. The majority of clinical data will be collected from electronic health records (EHRs) in order to reduce participant burden, while only a minority of the data will be collected through questionnaires, as outlined in table 2.

Furthermore, PRISMA consists of electronic Data Capture Systems (eDCSs), with each eDCS covering a certain theme, as described in more detail in table 3.

Data are registered via local data capture platforms, such as Castor EDC[20] (Ciwit, the Netherlands, which is hosted by True[21] in the Netherlands), and are centrally stored in Project Manager Internet Server,[22] a web-based relational database management system (Advanced Data Management, the Netherlands). These systems are compliant with Good Clinical Practice and are ISO27001 certified. All patients are being registered under a study number that is electronically assigned by a designated tool. This study number is used during data collection and data processing.

**Table 2** The PRISMA information model of the AAA bank

| eDCS name | Inclusion | | | Follow-up[N] | | | Stream | Source |
|---|---|---|---|---|---|---|---|---|
| | Asymptomatic AAA | Operation for acute AAA | Previously operated AAA | Surveillance phase | Surgical phase | Postoperative phase | | |
| Patient information | | 1 | 1 | | M | | | EHR |
| Informed consent | | 1 | | | M | | | n.a. |
| Living situation | | 1 | 1 | | 1 | | | Q |
| Current aneurysm state | | 1 | | | 1 | | | EHR |
| First occurrence comorbidity | | 1 | | | M | | | EHR |
| Family history | | 0.n | | | M | | | Q |
| Social history | | 1 | | | M | | | Q |
| Initial aneurysm characteristics | | 1 | | | | | | EHR |
| Comorbidity status | | 1 | | | 1 | | | EHR |
| Medication | | 1 | | | 1 | | | EHR, Q |
| Intoxications | | 1 | | | 1 | | | EHR, Q |
| Physical examination | | 1 | | 1 | 1 | | | EHR, Q |
| Blood test results | | 1 | | | 1 | | | EHR |
| Surveillance | 1 | | | 1 | | | | EHR |
| Preoperative assessment | | | 1 | | 1 | | | EHR |
| AAA repair | | | 1 | | 1 | | | EHR |
| Postoperative admission | | | 1 | | 1 | | | EHR |
| Postoperative follow-up | | | 1 | | | 1 | | EHR |
| Biomaterials | | 1 | | | 1 | | | n.a. |
| Cardiovascular events | | | | | | | dt | EHR |
| Other surgical procedures | | | | | | | dt | EHR |
| Malignancies | | | | | | | dt | EHR |
| AAA imaging | | 0.n | | | | | dt | EHR |
| Complications | | | | | | | dt | EHR |
| Imaging data | | 1 | | | 1 | | dt | EHR |

The eDCS from previous observation periods remain unchanged.

0.n: indicates that this eDCS can be registered multiple times (but is not mandatory).

1: Indicates that an eDCS is registered once and is overwritten in case of future changes.

AAA, abdominal aortic aneurysm; dt, time-stream with date-time format; eDCS, electronic Data Capture System; EHR: electronic health records; M, means 'Modify' and indicates that the eDCS can be modified within the same observation period. N: indicates that this eDCS can be registered multiple times (but at least once); PRISMA, Parelsnoer Repository for Information Specification, Modelling and Architecture; Q, questionnaire.

**Table 3** General description of each eDCS in PRISMA

| eDCS | General description |
|---|---|
| Patient information | Study number, year of birth and survival status including cause of death. |
| Informed consent | Status of informed consent (given or withdrawn), date of informed consent and individual patient decisions with regard to additional consent options. |
| Living situation | For example, independent or assisted living. |
| Current aneurysm state | For example, asymptomatic, acute or postoperative status. |
| First occurrence comorbidity | This eDCS contains a predefined list of comorbidities. Only if a participant has any of these comorbidities will this be registered, including the initial date of diagnosis. The list includes: hypertension, COPD, idiopathic lung fibrosis, peptic ulcer, diabetes mellitus, hypothyroidism and hyperthyroidism, hepatitis, portal hypertension, liver cirrhosis, any other liver disease, peripheral arterial disease, intracranial aneurysm, popliteal aneurysm, any other aneurysm, connective tissue disease, hypercholesterolaemia, carotid artery disease, renal dysfunction, renal disorders, ischaemic heart disease, heart failure, heart arrhythmia, heart valve disorders, any other cardiac disorder, haemiplegia or paraplegia, dementia and AIDS. Free-text descriptions can be provided for cases with an 'any other' comorbidity as mentioned above. |
| Family history | Registration of each known family member with (a history of) aortic aneurysmal disease and if known, at what age and whether there was an aortic rupture. |
| Social history | Marital status, employment status and educational status. |
| Initial aneurysm characteristics | Pathogenesis and type of AAA, status of AAA at time of inclusion, date of initial diagnosis and shape of AAA. |
| Comorbidity status | Status of a predefined list of comorbidities. This will be registered if this is known at the time of a visit to the department of vascular surgery. The list includes: COPD, diabetes mellitus (eg, dependency on insulin), peripheral arterial disease (eg, Fontaine classification), renal dysfunction, angina pectoris and heart failure. |
| Medication | This eDCS contains a predefined list of medications. Only pills and injections will be registered. If a medication is not included in the list, it will be added via a free-text description. Dosage will be registered for cardiovascular medications. The list includes: all types of cardiovascular medications, diabetes mellitus medication, corticosteroids, immunosuppressive medication and benzodiazepines. |
| Intoxications | Nicotine and alcohol use. |
| Physical examination | Body weight and patient length. |
| Blood test results | This eDCS contains a predefined list of blood tests, which will be registered if these have been carried out for clinical care. The list includes: creatinine, estimated glomerular filtration rate, total cholesterol, high-density lipoprotein cholesterol, low-density lipoprotein cholesterol and triglycerides, |
| Surveillance | Date of surveillance visit, treatment plan including motivation. |
| Preoperative assessment | American Society of Anaesthesiologists physical status classification |
| AAA repair | Date of AAA repair, type of AAA repair and intraoperative complications. |
| Postoperative admission | Length of hospitalisation and intensive care hospitalisation |
| Postoperative follow-up | Date of postoperative follow-up visit and treatment plan including motivation. |
| Cardiovascular events | Myocardial infarction, cerebrovascular accident and thromboembolic events including dates. |
| Other surgical procedures | This eDCS contains a predefined list of certain procedures. The list includes: cardiac surgery, vascular surgery, abdominal and thoracic surgery and transplant surgery. |
| Malignancies | General type of malignancy, date of diagnosis and type of treatments. |
| AAA imaging | Date and type of AAA imaging. AAA status at time of imaging (eg, asymptomatic, acute or postoperative) and AAA diameter. |

Continued

**Table 3** Continued

| eDCS | General description |
|---|---|
| Complications | Complications after any AAA repair will be registered according to a predefined code list of the Dutch National Surgical Complications Registry, which was established by a committee of the Association of Surgeons in the Netherlands. |

AAA, abdominal aortic aneurysm; COPD, chronic obstructive pulmonary disease; eDCS, electronic Data Capture Systems; PRISMA, Parelsnoer Repository for Information Specification, Modelling and Architecture.

### Biomaterials

Three types of biomaterials are collected: blood, urine and AAA tissue (table 4).

Blood and urine are collected repeatedly during the surveillance phase, up to a maximum of once per year to limit patient burden. Blood is saved as plasma, serum and whole blood for DNA and RNA. When open surgical repair of AAA is performed, aortic tissue is collected, snap frozen and stored at −80°C, and as formalin-fixed paraffin-embedded tissue. After surgery for AAA, blood and urine are collected up to 1 year postoperatively (table 4). All procedures concerning biomaterials adhere to standard operating procedures outlined by PSI. Furthermore, all biomaterials are stored in designated PSI biobanks within each UMC. Thus, all biomaterials are stored in the UMC where they are collected.

### Imaging data

In order to facilitate future imaging studies, CT and MRI data that are obtained as part of clinical care are also stored centrally. At the time of writing, the participating centres are harmonising the CT protocols to acquire standardised images. Partners currently involved in the central storage of images are Translational Research IT (TRaIT; part of Dutch non-profit organisation Lygature) and Vancis, a Dutch service provider of IT services for research with a ISO27001 certificate.[23–25] The imaging data are stored centrally in an Extensible Neuroimaging Archive Toolkit (XNAT) server (Buckner Lab, Harvard University, USA).[26] This server is operated by TraIT and is hosted by Vancis.

Imaging data are stored as Digital Imaging and Communications in Medicine (DICOM) data. Before the data are sent to the central server, the study number is allocated to the imaging data to enable linking of imaging data with clinical data and biomaterials. All other identifiable data are removed from the DICOM data using Clinical Trial Processor (Radiological Society of North America, USA).[27]

### Statistical analyses

The primary outcomes are AAA growth, AAA rupture and all-cause mortality. AAA growth will be based on multiple measurements and will therefore be analysed with a repeated measures analysis. Potential associations between AAA growth and risk factors that are also measured on multiple time points can be assessed with multivariable mixed-effects models, while potential associations between AAA rupture and risk factors can be tested with a conditional dynamic prediction model with landmarking, or with joint models in which linear mixed-effects models are combined with Cox regression.

Due to the expected multifactorial aspect of AAA growth, even weak correlations are of interest to detect.

**Table 4** Biomaterials collected for the AAA bank

| Biomaterials | Inclusion | Surveillance phase | Surgical phase | Postoperative phase | Storage temperature |
|---|---|---|---|---|---|
| Blood: EDTA plasma | Once | Max. 1/year | Once | Once: 1 year postoperative | ≤−80°C |
| Blood: serum | Once | Max. 1/year | Once | Once: 1 year postoperative | ≤−80°C |
| Blood: citrate plasma | Once | Max. 1/year | Once | Once: 1 year postoperative | ≤−80°C |
| Blood: EDTA for DNA | Once | | | | 4°C or ≤−20°C |
| Blood: PAXgene for RNA | Once | | | | ≤−80°C |
| Urine | Once | Max. 1/year | Once | Once: 1 year postoperative | ≤−80°C |
| Aneurysm tissue: frozen | | | Once | | ≤−80°C |
| Aneurysm tissue: FFPE | | | Once | | Room temperature |

FFPE, formalin-fixed paraffin-embedded.

| Table 5 | The PARIS study, the first proposed study that will be facilitated by the AAA bank |
|---|---|
| Aims | The aims of the PARIS study are: (1) to determine the association between AAA progression (growth or rupture) and the evolution of serum levels of both proteases and cytokines, (2) to determine the association between overall survival and the evolution of serum levels of proteases and cytokines, (3) to determine the association between serum levels of proteases and cytokines and level of proteases and cytokines in AAA tissue and finally (4) to determine the incidence of and characterise the type of complications after AAA repair. |
| Outcomes | The primary outcomes are: AAA growth, AAA rupture, all-cause mortality, serum levels of cytokines and proteases and cytokine and protease levels in AAA tissue. Secondary outcomes are the incidence and type of complications after AAA repair. |
| Sample size | Because of the multifactorial aspect of AAA progression, we calculated a sample size that will be sufficient for detecting weak correlations between cytokine and protease levels and AAA growth. To detect a correlation coefficient of 0.16 with a power of 80%, and a significance level of. 05, a sample size of 750 participants is required. Strategies to achieve sufficient participant enrolment include simple eligibility criteria and the expressed intention to extend the number of participating centres. |
| Statistical analysis | Categorical variables will be presented as numbers and percentages and will be compared between groups with the $\chi^2$ test of Fisher's exact test where appropriate. Continuous variables will be presented as means±SD or as medians with the IQR, depending on distribution. Distribution of continuous variables will be tested with the Shapiro-Wilk test. Continuous variables will be compared between groups with either the unpaired t-test or the Mann-Whitney U test depending on distribution. |
| | Individual AAA growth (based on repeated AAA diameter measurements) and the evolution of serum levels of cytokines and proteases will each be analysed with linear mixed-effects models to estimate the slope of temporal change (ie, the evolution through time). The linear mixed-effects model that estimates the slope for the evolution of each biomarker will also adjust for potentially relevant covariables including but not limited to sex, age and cardiovascular comorbidity. A forward stepwise selection will be used to select only those covariables that were significant in univariable analysis. Finally, correlations coefficient will be estimated between the slope of AAA progression and the slope of each biomarker. |
| | Freedom from AAA rupture and all-cause mortality will be estimated with the Kaplan-Meier method, while joint modelling will be used to assess the association between the evolution of serum levels of proteases and cytokines and AAA rupture and all-cause mortality. A joint model will be performed per biomarker. In order to assess the association, joint modelling combines linear mixed-effects models for the estimation of slope of temporal change per biomarker with Cox regression for the analysis of freedom from AAA rupture and all-cause mortality, in order to estimate HRs. The estimated slope and value of the studied biomarker will be added as covariables in each joint model. Furthermore, both models will be adjusted for potentially relevant covariables, including but not limited to sex, age, most recent AAA diameter and cardiovascular comorbidity using the same selection procedure. In addition, AAA growth instead of the most recent AAA diameter will be analysed in a separate multivariable joint model. In all analyses, a p value of <0.05 will be considered statistically significant, including the covariable selection procedure. |

AAA, abdominal aortic aneurysm; PARIS, Predicting aneurysm growth and rupture with longitudinal biomarkers.

To that end, at least 750 patients are required to be able to detect a correlation coefficient of 0.16 with a power of 80% and a significance level of 0.05.

## Patient and public involvement

This research was done without patient involvement. Patients were not invited to comment on the study design and were not consulted to develop patient relevant outcomes or interpret the results. Patients were not invited to contribute to the writing or editing of this document for readability or accuracy.

## ETHICS AND DISSEMINATION

The Medical Ethics Board and the Biobank Ethics Board of Amsterdam UMC (University of Amsterdam) have approved the AAA bank together with the PARIS study (table 5) within the scope of the Dutch Medical Research Involving Human Subjects Act (WMO) under registration number NL59991.018.17.

In general, biobanks in the Netherlands do not fall within the scope of the WMO. To be eligible for assessment under WMO, the formulation of a specific research question is required. However, specific research questions are often not present at the moment of biobank initiation.[28] Yet, the submission of the AAA bank together with the PARIS study (that contains a specific research question) enabled approval of the combined project within the scope of the WMO. This approval ensures that the AAA bank adheres to the highest legal and medical ethical standards and that participation of other future centres can be realised using the existing procedures of the WMO. Because of this design, all participating patients sign two informed consent forms—one for the biobank and one for the PARIS study (see online appendix 1–6 for English versions of the forms). By consenting to the AAA bank and signing its consent form, patients consent to the collection and storage of their biomaterials and data and to future analyses of it for research about AAA.

Any significant modification to the protocol that may impact patient safety or the conduct, design or analysis of the study requires formal amendment to the protocol. These will need to be approved by the Medical Ethics Board and the Biobank ethics Board of Amsterdam UMC (University of Amsterdam).

To accommodate patients with different views on data collection, participants can refuse collection of DNA and the sharing of their data with foreign and/or commercial parties. All collected data and biomaterials will be stored for a maximum of 50 years. When a participant decides to withdraw from the AAA bank, all stored biomaterials and data will be destroyed or deleted. When reasonably possible, this is also done with materials that are sent out for a specific study.

### Scientific board for future studies

Collected data and biomaterials of the AAA bank can only be used for future studies that fall within the scope of the scientific aims of the AAA bank and that are approved by the scientific board. Researchers can submit a study proposal with the scientific board of the AAA bank. Each study proposal must include a study objective and/or research questions, the type of data and biomaterials required, a statistical analysis plan and an agreed on authorship policy. This board oversees all requests for data and consists of five members, with a minimum of one biostatistician (A H Zwinderman, Amsterdam UMC, University of Amsterdam, Department of Clinical Epidemiology) and either a legal expert or ethicist (provided by PSI) among its members. The other three members are currently vascular surgeons from the initiating UMCs (DL, WW and JH). If a study proposal is approved by the scientific board, subsequent medical ethical approval will be acquired if required by Dutch law or local guidelines. Furthermore, data are only released in accordance with standard PSI procedures.

Results of individual studies will be published in peer-reviewed scientific journals and will be presented at international conferences.

**Acknowledgements** We would like to thank Parelsnoer Institute for their help in initiating and designing the abdominal aortic aneurysm (AAA) bank and the Predicting aneurysm growth and rupture with longitudinal biomarkers (PARIS) study. In addition, we would like to thank Roger Snijder, information architect of Parelsnoer Institute (PSI), for his help in designing our information model.

**Contributors** HJ and RI contributed equally to this manuscript. All authors actively contributed to the study conception, design and its protocol. All authors read and contributed to the writing of the manuscript and approved the final manuscript.

**Funding** This research received no specific grant from any funding agency in the public, commercial or not-for-profit sectors.

**Competing interests** None declared.

**Provenance and peer review** Not commissioned; externally peer reviewed.

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
