## [Reviewer comments · BMJ Open]

ARTICLE DETAILS

TITLE (PROVISIONAL)	Design and protocol of a comprehensive multicentre biobank for abdominal aortic aneurysms
AUTHORS	Jalalzadeh, Hamid; Indrakusuma, Reza; Blankensteijn, Jan; Wisselink, Willem; Yeung, Kakkhee; Lindeman, Jan; Hamming, Jaap; Koelemay, Mark; Legemate, Dink; Balm, Ron

VERSION 1 – REVIEW

REVIEWER	Wing Cheuk Chan Counties Manukau District Health Board, New Zealand
REVIEW RETURNED	02-Feb-2019

GENERAL COMMENTS	The development of a biobank that includes clinical, imaging and biological data has the potential to improve understanding of the pathophysiology and natural progression of abdominal aortic aneurysm (AAA). The protocol has a number of strengths: It includes a longitudinal coverage of clinical information that include serum, urine, and AAA tissue and some relevant AAA outcomes. A comprehensive range of biomaterial including DNA is collected. Multiple centres are expected to run the same protocol for CT imaging that would enable better comparability between different centres and over time. However, protocol can be further strengthened by a clarifying a number of considerations below and adding causes of death as an important outcome to be captured which is highly feasible to do so. 1. Since people with AAA often have competing co-morbidities,¹ and more likely to die from other causes rather than from AAA rupture in particular from people under AAA surveillance, the protocol would be greatly enhanced if there is a standardised way to capture relevant risk factors, co-morbidities, and the corresponding severity in a consistent and systematic way. It would be helpful to provide more detail on the risk factors (e.g. smoking history) and co-morbidities are expected to be captured (as well as cardiovascular disease and malignancy). The future results would be much more interpretable if risk factors and co-morbidities and the corresponding severity are better captured consistently, e.g. are certain biomarkers independent predictors of AAA progression or deaths caused by AAA?2. Since survival from AAA rupture has improved significantly over time, it would be critical to specify the causes of death as an outcome (along with all cause mortality). This should be feasible as causes of death are routinely captured in the Netherlands.3. It would be worthwhile clarify if table 1 and figure 1 are internally consistent in terms of logic. Table 1 states inclusion criteria is adult with AAA or who had been previously treated for
---

	AAA. However, not all patients who have an asymptomatic AAA would benefit from or indicated for surveillance. For example, people with severe co-morbidities who are clearly not eligible from endovascular intervention. Are they excluded from the AAA bank? If so update table 1 accordingly, if not update figure 1 flow chart to include a category who are not eligible for therapy, and any followup scans would be of incidental in nature as opposed to active surveillance. Minor comments Introduction: page 7 line 14 that states “management of patients with a AAA is aimed at preventing rupture” is a vascular intervention team centric viewpoint rather than a patient centric or population centric viewpoint. Considering the EVAR 2 trial demonstrated that for people with AAA (≥5.5cm) who were not eligible for open repair because of co-morbidities, endovascular repair reduced aneurysm related mortality without increasing overall survival,² preventing AAA rupture alone for some people may not translate to any improvement in either quality or quantity of life. Many people with large AAA may not die from AAA rupture.³ Therefore, it may be more appropriate to state “management of patients with a AAA should be aiming to improve the patients’ quantity and quality of life” rather than preventing rupture alone. Overall recommendation: Consider the potential importance of the databank to be more widely known to a range of researchers, suggest acceptance if the above points are adequately addressed. References  1. Guirguis-Blake JM, Beil TL, Senger CA, Whitlock EP. Ultrasonography screening for abdominal aortic aneurysms: a systematic evidence review for the U.S. Preventive Services Task Force. Ann Intern Med 2014; 160(5): 321-9. 2. Sweeting MJ, Patel R, Powell JT, Greenhalgh RM, Investigators ET. Endovascular Repair of Abdominal Aortic Aneurysm in Patients Physically Ineligible for Open Repair: Very Long-term Follow-up in the EVAR-2 Randomized Controlled Trial. Ann Surg 2017; 266(5): 713-9. 3. Scott SW, Batchelder AJ, Kirkbride D, Naylor AR, Thompson JP. Late Survival in Nonoperated Patients with Infrarenal Abdominal Aortic Aneurysm. Eur J Vasc Endovasc Surg 2016; 52(4): 444-9.
--	---

REVIEWER	Alan Batterham Teesside University, UK.
REVIEW RETURNED	03-Feb-2019

GENERAL COMMENTS	The AAA bank is an excellent and important initiative and the data sharing – subject to the scientific quality of the request – will facilitate substantial research in this field internationally. The protocol is described well, and I have just a few comments/ observations.  1. I could not find any simple measures of body size mentioned in the protocol, though I assume that height and body mass are collected routinely. I raise this issue, as there is an opportunity here with large datasets to learn much more about how AAA size scales to general body size. This knowledge has implications for
--

	clinical decision making, especially when combined with other data. Dewey et al. (2008) highlight this issue in one of their clinical vignettes, describing a 66 year-old woman 1.55 m tall and 49 kg body mass, with a 50 mm aneurysm. Although this woman is in a grey zone between surgery and watchful waiting, her aneurysm size is actually large for her relatively small body size according to what we know currently about the allometric scaling of AAA size to body size. I can envisage data sharing applications in this area in due course. Dewey FE, Rosenthal D, Murphy Jr DJ, Froelicher VF, Ashley EA. Does Size Matter? Clinical Applications of Scaling Cardiac Size and Function for Body Size. Circulation. 2008;117(17):2279-87. Doi 10.1161/CIRCULATIONAHA.107.736785. 2. This is a minor point, but please change 'multivariate' to 'multivariable', when you are referring to multiple predictor variables. 3. Line 46. "...have not corrected for factors as smoking..." Insert 'such', as in 'such as'. 4. The scientific aims are appropriate and it is good to see that future studies using data from the AAA bank must be conducted in accordance with these aims. 5. It is pleasing to see that all measures are embedded within routine clinical care, with no additional burden to patients from the research. 6. The description of the statistical analyses in the protocol is very brief. I appreciate that it is not possible to be specific, as many different studies and analyses might be performed using data from the AAA bank. I presume that detailed statistical analysis plans will be required for any AAA bank study – is there one for the PARIS study that readers could be directed to?
--	--

VERSION 1 – AUTHOR RESPONSE

Reviewer(s)' Comments to Author:

Reviewer: 1

Reviewer Name: Wing Cheuk Chan

Institution and Country: Counties Manukau District Health Board, New Zealand Please state any competing interests or state 'None declared': None Declared.

Thank you for reviewing this manuscript.

Please leave your comments for the authors below

The development of a biobank that includes clinical, imaging and biological data has the potential to improve understanding of the pathophysiology and natural progression of abdominal aortic aneurysm (AAA). The protocol has a number of strengths: It includes a longitudinal coverage of clinical information that include serum, urine, and AAA tissue and some relevant AAA outcomes. A comprehensive range of biomaterial including DNA is collected. Multiple centres are expected to run

the same protocol for CT imaging that would enable better comparability between different centres and over time.

However, protocol can be further strengthened by a clarifying a number of considerations below and adding causes of death as an important outcome to be captured which is highly feasible to do so.

1. Since people with AAA often have competing co-morbidities,¹ and more likely to die from other causes rather than from AAA rupture in particular from people under AAA surveillance, the protocol would be greatly enhanced if there is a standardised way to capture relevant risk factors, co-morbidities, and the corresponding severity in a consistent and systematic way. It would be helpful to provide more detail on the risk factors (e.g. smoking history) and co-morbidities are expected to be captured (as well as cardiovascular disease and malignancy). The future results would be much more interpretable if risk factors and co-morbidities and the corresponding severity are better captured consistently, e.g. are certain biomarkers independent predictors of AAA progression or deaths caused by AAA?

We agree with you that the registration of comorbidities is of the utmost importance, yet it can be challenging to register in longitudinal cohorts without overly burdening participants. This is why our data collection primarily relies on secondary collection of data from electronic health records (e.g. comorbidity data), which are supplemented with data from participant questionnaires (e.g. smoking history, alcohol use, social history and actual medication use). We have revised the methods to reflect on this. We have also included a new table (table 3) that gives additional details regarding what data we aim to collect. This includes smoking history, cardiovascular disease and malignancies.

Accurately registering comorbidity severity is another challenging aspect in a longitudinal cohort that regards minimising participant burden as a priority. Although our information model contains an eDCS concerning comorbidity severity, or rather, its current status, this is only limited to a few selected comorbidities as shown in the new table.

Revised text in the paragraph “Study procedures” in the methods section (page 10, marked copy):
“The study procedures of the AAA bank are embedded within regular AAA treatment and are carried out by physician-researchers and data managers. Standard operating procedures have been set up to minimise the amount of missing data.”

Revised text in the paragraph “Clinical data – PRISMA” in the methods section (page 11, marked copy):

“All clinical data are collected according to an information model called Parelsnoer Repository for Information Specification, Modelling and Architecture (PRISMA), which was constructed with the assistance of an experienced information architect of PSI. The majority of clinical data will be collected from electronic health records (EHRs) in order to reduce participant burden, while only a minority of the data will be collected through questionnaires, as outlined in table 2. Furthermore, PRISMA consists out of electronic Data Capture Systems (eDCSs), with each eDCS covering a certain theme, as described in more detail in Table 3.

Data are registered via local data capture platforms, such as Castor EDC[20] (Ciwit, the Netherlands, which is hosted by True[21] in the Netherlands), and are centrally stored in Project Manager Internet Server (ProMISe)[22], a web-based relational database management system (Advanced Data Management, the Netherlands). These systems are compliant with Good Clinical Practice and are ISO27001 certified. All patients are being registered under a study number that is electronically assigned by a designated tool. This study number is used during data collection and data processing.”

2. Since survival from AAA rupture has improved significantly over time, it would be critical to specify the causes of death as an outcome (along with all cause mortality). This should be feasible as causes of death are routinely captured in the Netherlands.

We have clarified that we register cause of death in the new table 3 (page 15, marked copy).

3. It would be worthwhile clarify if table 1 and figure 1 are internally consistent in terms of logic. Table 1 states inclusion criteria is adult with AAA or who had been previously treated for AAA. However, not all patients who have an asymptomatic AAA would benefit from or indicated for surveillance. For example, people with severe co-morbidities who are clearly not eligible from endovascular intervention. Are they excluded from the AAA bank? If so update table 1 accordingly, if not update figure 1 flow chart to include a category who are not eligible for therapy, and any followup scans would be of incidental in nature as opposed to active surveillance.

Thank you for this observation. Indeed, our inclusion criteria allow the inclusion of patients with an asymptomatic AAA irrespective of the treatment after inclusion. Patients are not excluded based on their expected follow-up duration of treatment. Consequently, we include patients for whom it is, or becomes, clear that any type of AAA repair will not be beneficial. In such cases, the decision to continue with surveillance is made together with patients themselves, because this decision is multifactorial. Some patients continue their surveillance in which case it still is active surveillance (and not incidental), while others choose to quit with surveillance altogether and do not come back to the department of vascular surgery (i.e. they will be lost to follow-up). In all cases, participation in the AAA bank continues so long as participants are seen at the department of vascular surgery. We have revised this in the text.

Revised text at the end of paragraph "Study procedures" in the methods section (page 11, marked copy):

"Furthermore, there will be patients who at some point reach the threshold diameter for repair, yet who do not undergo repair for various reasons such as severe comorbidity. In these cases, an individual clinical decision – unrelated to their participation with the AAA bank – will have to be made together with the patient whether surveillance continues with regular intervals or whether the patient chooses to quit with surveillance altogether. Patients who choose to continue with surveillance will still be asked for biomaterials and clinical data, while patients who quit surveillance will be considered lost to follow-up in future analyses."

Minor comments

Introduction: page 7 line 14 that states "management of patients with a AAA is aimed at preventing rupture" is a vascular intervention team centric viewpoint rather than a patient centric or population centric viewpoint. Considering the EVAR 2 trial demonstrated that for people with AAA (≥ 5.5 cm) who were not eligible for open repair because of co-morbidities, endovascular repair reduced aneurysm related mortality without increasing overall survival,² preventing AAA rupture alone for some people may not translate to any improvement in either quality or quantity of life. Many people with large AAA may not die from AAA rupture.³ Therefore, it may be more appropriate to state "management of patients with a AAA should be aiming to improve the patients' quantity and quality of life" rather than preventing rupture alone.

We agree with you that this line suggests a rather narrow perspective on our part.

Revised text in the introduction (page 5, marked copy):

"Yet, the risk of rupture itself is difficult to measure accurately and also varies considerably between patients. Consequently, the management of patients with an asymptomatic AAA is focused on balancing the risk of rupture with other competing risks of death, with the aim of preserving quality

and quantity of life. On the one hand, asymptomatic patients who are estimated to have a high risk of rupture, in general at a diameter of more than 5.5 cm for men, may be offered prophylactic AAA repair if the risk of rupture outweighs any procedural and/or competing risks.[4] On the other hand, asymptomatic patients for whom the risk of rupture is estimated to be smaller than procedural and/or competing risks will be offered surveillance – for example those with an AAA diameter smaller than 5.5 cm or those with severe comorbidities.”

Overall recommendation:

Consider the potential importance of the databank to be more widely known to a range of researchers, suggest acceptance if the above points are adequately addressed.

References

1. Guirguis-Blake JM, Beil TL, Senger CA, Whitlock EP. Ultrasonography screening for abdominal aortic aneurysms: a systematic evidence review for the U.S. Preventive Services Task Force. *Ann Intern Med* 2014; 160(5): 321-9.
2. Sweeting MJ, Patel R, Powell JT, Greenhalgh RM, Investigators ET. Endovascular Repair of Abdominal Aortic Aneurysm in Patients Physically Ineligible for Open Repair: Very Long-term Follow-up in the EVAR-2 Randomized Controlled Trial. *Ann Surg* 2017; 266(5): 713-9.
3. Scott SW, Batchelder AJ, Kirkbride D, Naylor AR, Thompson JP. Late Survival in Nonoperated Patients with Infrarenal Abdominal Aortic Aneurysm. *Eur J Vasc Endovasc Surg* 2016; 52(4): 444-9.

Reviewer: 2

Reviewer Name: Alan Batterham

Institution and Country: Teesside University, UK.

Please state any competing interests or state 'None declared': None declared.

Thank you for reviewing this manuscript.

Please leave your comments for the authors below The AAA bank is an excellent and important initiative and the data sharing – subject to the scientific quality of the request – will facilitate substantial research in this field internationally. The protocol is described well, and I have just a few comments/ observations.

1. I could not find any simple measures of body size mentioned in the protocol, though I assume that height and body mass are collected routinely. I raise this issue, as there is an opportunity here with large datasets to learn much more about how AAA size scales to general body size. This knowledge has implications for clinical decision making, especially when combined with other data. Dewey et al. (2008) highlight this issue in one of their clinical vignettes, describing a 66 year-old woman 1.55 m tall and 49 kg body mass, with a 50 mm aneurysm. Although this woman is in a grey zone between surgery and watchful waiting, her aneurysm size is actually large for her relatively small body size according to what we know currently about the allometric scaling of AAA size to body size. I can envisage data sharing applications in this area in due course.

We agree with you that body size can be of importance for the reason you have outlined. We register body weight and patient length via questionnaires at a maximum frequency of once yearly. This is to register baseline values and also any potential changes. We have clarified this in the manuscript by adding an extra table (table 3, page 15, marked copy) that gives additional explanation to the information model, and by highlighting the source of data in table 2 (page 13, marked copy).

Dewey FE, Rosenthal D, Murphy Jr DJ, Froelicher VF, Ashley EA. Does Size Matter? Clinical Applications of Scaling Cardiac Size and Function for Body Size. *Circulation*. 2008;117(17):2279-87. Doi 10.1161/CIRCULATIONAHA.107.736785.

2. This is a minor point, but please change 'multivariate' to 'multivariable', when you are referring to multiple predictor variables.

We have revised this.

3. Line 46. "...have not corrected for factors as smoking..." Insert 'such', as in 'such as'.

Thank you for spotting this error, we have revised this.

4. The scientific aims are appropriate and it is good to see that future studies using data from the AAA bank must be conducted in accordance with these aims.

5. It is pleasing to see that all measures are embedded within routine clinical care, with no additional burden to patients from the research.

Thank you, we agree that minimising participant burden is important.

6. The description of the statistical analyses in the protocol is very brief. I appreciate that it is not possible to be specific, as many different studies and analyses might be performed using data from the AAA bank. I presume that detailed statistical analysis plans will be required for any AAA bank study – is there one for the PARIS study that readers could be directed to?

You are correct that all study proposal submitted with the AAA bank will require a statistical analysis plan. We have revised the text to clarify this.

Revised text in the paragraph "Scientific board for future studies" in the methods section (page 25, marked copy):

"Each study proposal must include a study objective and/or research questions, the type of data and biomaterials required, a statistical analysis plan and an agreed upon authorship policy."

Furthermore, we have included a more detailed statistical analysis plan for the PARIS study in table 5 (page 23, marked copy).

VERSION 2 – REVIEW

REVIEWER	Wing Cheuk Chan Counties Manukau District Health Board, New Zealand
REVIEW RETURNED	31-Mar-2019

GENERAL COMMENTS	Design and protocol of a comprehensive multicentre biobank for abdominal aortic aneurysms The authors have answered most of my comments from my previous review adequately. The additional tables are useful. Thank you. In the process of clarifying inclusive and exclusion criteria of participants, an important potential bias might have been inadvertently introduced and merit further clarification. "Patients who quit surveillance will be considered lost to follow-up in future analyses."
---

	Since substantial proportions of people with AAA do not die from an AAA rupture, many people might longer be under surveillance or under the care of vascular surgeons prior to death. A cohort of people who did not receive AAA interventions or with slow AAA progression or growth might not be analysed because of this pre-defined approach to further analyses. It may be more helpful to acknowledge that people who are no longer under surveillance will not have some (or all) of clinical, imaging or biomaterial recorded in the biobank. However, these cases can still be analysed in regard to important outcomes such as mortality as a sensitivity analyses. The fact of death and cause of death information could be sourced from the national mortality collections as well as from the electronic health records of primary care to ensure better completeness?
--	---

REVIEWER	Alan Batterham Teesside University, UK.
REVIEW RETURNED	28-Mar-2019

GENERAL COMMENTS	I am happy with the authors' responses and associated edits to the paper.
---

VERSION 2 – AUTHOR RESPONSE

Reviewer: 1

Reviewer Name: Wing Cheuk Chan

Institution and Country: Counties Manukau District Health Board, New Zealand

Please state any competing interests or state 'None declared': None Declared

Please leave your comments for the authors below

Design and protocol of a comprehensive multicentre biobank for abdominal aortic aneurysms

The authors have answered most of my comments from my previous review adequately. The additional tables are useful. Thank you. In the process of clarifying inclusive and exclusion criteria of participants, an important potential bias might have been inadvertently introduced and merit further clarification.

“Patients who quit surveillance will be considered lost to follow-up in future analyses.”

Since substantial proportions of people with AAA do not die from an AAA rupture, many people might longer be under surveillance or under the care of vascular surgeons prior to death. A cohort of people who did not receive AAA interventions or with slow AAA progression or growth might not be analysed because of this pre-defined approach to further analyses.

It may be more helpful to acknowledge that people who are no longer under surveillance will not have some (or all) of clinical, imaging or biomaterial recorded in the biobank. However, these cases can still be analysed in regard to important outcomes such as mortality as a sensitivity analyses. The fact of

death and cause of death information could be sourced from the national mortality collections as well as from the electronic health records of primary care to ensure better completeness?

Thank you for your additional review and highlighting this unintended consequence of our revision. We would like to clarify that conservatively managed patients who withdraw from surveillance will be analysed using the data that has already been collected, additionally augmented by mortality data from other sources.

Revised text in that paragraph (page 11 main document – marked copy):

“Patients who choose to continue with surveillance will still be asked for biomaterials and clinical data, while patients who quit surveillance can be analysed using the previously collected data up until that moment. Furthermore, the latter patients can also be included in survival analyses as mortality data can be sourced from either the municipal registry of persons or their general practitioner.”

Reviewer: 2

Reviewer Name: Alan Batterham

Institution and Country: Teesside University, UK.

Please state any competing interests or state ‘None declared’: None declared.

Please leave your comments for the authors below

I am happy with the authors' responses and associated edits to the paper.

Thank you for your additional review.